# Immunohistochemistry-Based Taxonomical Classification of Bladder Cancer Predicts Response to Neoadjuvant Chemotherapy

**DOI:** 10.3390/cancers12071784

**Published:** 2020-07-03

**Authors:** Albert Font, Montserrat Domènech, Raquel Benítez, Marta Rava, Miriam Marqués, José L. Ramírez, Silvia Pineda, Sara Domínguez-Rodríguez, José L. Gago, Josep Badal, Cristina Carrato, Héctor López, Ariadna Quer, Daniel Castellano, Núria Malats, Francisco X. Real

**Affiliations:** 1Medical Oncology Department, Institut Català d'Oncologia, Badalona Applied Research group in Oncology (B-ARGO Group), University Hospital Germans Trias I Pujol, 08916 Badalona, Spain; afont@iconcologia.net (A.F.); jramirez@iconcologia.net (J.L.R.); 2Medical Oncology Department, Fundació Althaia, 08243 Manresa, Spain; mdomenech@althaia.cat; 3Genetic and Molecular Epidemiology Group, Spanish National Cancer Research Centre (CNIO), 28029 Madrid, Spain; rbenitezd@ext.cnio.es (R.B.); rava.marta@gmail.com (M.R.); spineda@cnio.es (S.P.); sara.dominguez.r@gmail.com (S.D.-R.); 4Centro de Investigación Biomédica en Red de Cáncer (CIBERONC), 28029 Madrid, Spain; mmarques@cnio.es; 5Epithelial Carcinogenesis Group, Spanish National Cancer Research Centre (CNIO), 28029 Madrid, Spain; 6IGTP-Molecular Biology Laboratory, Institut Germans Trias i Pujol, 08916 Badalona, Spain; 7Urology Department, Hospital Universitari Germans Trias i Pujol, 08916 Badalona, Spain; zhigago@gmail.com; 8Pathology Department, Fundació Althaia, 08243 Manresa, Spain; jbadal@althaia.cat; 9Pathology Department, Hospital Germans Trias i Pujol, 08916 Badalona, Spain; ccarrato@hotmail.com (C.C.); ariadnaqs@gmail.com (A.Q.); 10Universitat Autònoma de Barcelona, 08193 Barcelona, Spain; 11Urology Department, Fundació Althaia, 08243 Manresa, Spain; hlopez@althaia.cat; 12Medical Oncology Department, Hospital 12 de Octubre, 28029 Madrid, Spain; cdanicas@hotmail.com; 13Departament de Ciències Experimentals i de la Salut, Universitat Pompeu Fabra, 08193 Barcelona, Spain

**Keywords:** bladder cancer, neoadjuvant chemotherapy, molecular taxonomy, immunohistochemistry, basal/squamous-like tumors

## Abstract

Background: Platinum-based neoadjuvant chemotherapy (NAC) increases the survival of patients with organ-confined urothelial bladder cancer (UBC). In retrospective studies, patients with basal/squamous (BASQ)-like tumors present with more advanced disease and have worse prognosis. Transcriptomics-defined tumor subtypes are associated with response to NAC. Aim: To investigate whether immunohistochemical (IHC) subtyping predicts NAC response. Methods: Patients with muscle-invasive UBC having received platinum-based NAC were identified. Tissue microarrays were used to type tumors for KRT5/6, KRT14, GATA3, and FOXA1. Outcomes: progression-free survival and disease-specific survival; univariable and multivariate Cox regression models were applied. Results: We found a very high concordance between mRNA and protein expression. Using IHC-based hierarchical clustering, we classified 126 tumors in three subgroups: BASQ-like (FOXA1/GATA3 low; KRT5/6/14 high), Luminal-like (FOXA1/GATA3 high; KRT5/6/14 low), and mixed-cluster (FOXA1/GATA3 high; KRT5/6 high; KRT14 low). Applying multivariable analyses, patients with BASQ-like tumors were more likely to achieve a pathological response to NAC (OR 3.96; *p* = 0.017). The clinical benefit appeared reflected in the lack of significant survival differences between patients with BASQ-like and luminal tumors. Conclusions: Patients with BASQ-like tumors—identified through simple and robust IHC—have a higher likelihood of undergoing a pathological complete response to NAC. Prospective validation is required.

## 1. Introduction

Cisplatin-based neoadjuvant chemotherapy (NAC) is the recommended treatment for locally advanced muscle-invasive bladder cancer (MIBC), with the highest level of evidence [1]. Meta-analyses have shown that the benefit of NAC is limited to a subset of patients, with an overall improvement of 5–6.5% in 5-year survival compared to cystectomy alone [2,3]. A recent meta-analysis shows a significant overall survival (OS) benefit associated with cisplatin-based NAC (hazard ratio (HR), 0.87; 95% confidence interval (CI), 0.79–0.96) [4]. Pathological complete response (pCR) or down-staging to non-MIBC (pT1-pTis) occur in almost half of patients treated with NAC. Patients who benefit the most from NAC are those achieving a pCR at cystectomy [4,5]. The inability to select patients who can benefit has limited the use of NAC: treatment can lead to unnecessary toxicity in patients who fail to respond, and it delays a potentially curative cystectomy, with a negative survival impact [6,7]. In fact, a recent study concluded that while patients who attain a pCR have longer survival, those who do not attain a pCR have a worse prognosis than patients who proceed directly to radical cystectomy without NAC [8]. These issues have reduced the use of NAC, highlighting the need of biomarkers to identify patients likely to benefit from NAC.

A better understanding of the molecular heterogeneity of urothelial bladder cancer (UBC) [9,10,11] has enabled classifying MIBC patients according to molecular profiles. Several groups have proposed classifications of all [12], non muscle-invasive bladder cancer (NMIBC) [13] or MIBC [9,10,11,14,15] tumors according to their transcriptome. Common to them is the identification of a tumor subgroup, resembling basal-like breast cancers and lacking urothelial luminal markers. There is consensus that basal/squamous-like (BASQ) tumors can be defined using four markers: FOXA1, GATA3, KRT5/6, and KRT14. FOXA1 and GATA3 are expressed throughout the urothelium, and they are involved in the expression of the urothelial differentiation program. In contrast, KRT14 is exclusively expressed in a subset of cells in the basal layer and KRT5 is expressed in basal/suprabasal layers, but it is excluded from the most luminal layers. Using these markers, BASQ tumors have been shown to be characterized as expressing KRT5/6 and KRT14 and lacking GATA3 and FOXA1 [16].

This classification is potentially relevant from a clinical standpoint as several studies have shown that patients with BASQ-type UBC have a more advanced disease at presentation and worse prognosis [14,17,18]. However, there is also evidence from retrospective studies that these tumors may respond better to NAC [14,18,19]. The BASQ subtype was originally reported using global transcriptomics but its phenotype can be recognized with high accuracy using immunohistochemistry (IHC) [20,21]. 

In this retrospective study, we have assessed the relationship between MIBC tumor subtypes— identified by IHC—and the response to NAC and survival. In complementary analyses, we also examined the agreement of the results of IHC and mRNA expression. The first molecular taxonomy classification of MIBC did not include markers of luminal tumors [16], although the KRT20 expression and alterations in several genes—including *FGFR3*, *STAG2*, and *PIK3CA*—have been postulated as molecular markers of this molecular subtype [22]. It remains unclear, however, whether combining the analysis of urothelial/luminal differentiation markers with markers of basal MIBC could improve its predictive capacity. In addition, oncogene mutations have been described in MIBC [10,11]. Therefore, we included them in order to explore whether they have an added value in the prediction of response to NAC. 

## 2. Results

### 2.1. Clinical-Pathological Features of Patients

This study is based on a retrospective series of patients with MIBC, treated with platinum-based NAC in two Spanish hospitals from 1994 to 2014 (n = 215). A fraction of patients was not analyzed; the main reason for patient exclusion was that the tumor tissue was unavailable (n = 76) because one of the hospitals was a referral center for other hospitals in its catchment area and the transurethral resection of the bladder tumor (TURBT) was performed elsewhere. To assess the possible selection biases, we compared the clinical-pathological characteristics between the groups of eligible vs. ineligible patients, and did not find strong evidence for the existence of major biases (Appendix A). An additional seven patients were not included because of the lack of information on all four IHC markers, not allowing cluster definition. Finally, a total of 126 patients were finally assessed.

The clinical-pathological characteristics of the patients from whom tumor tissue was available and those from whom no material was available were similar, indicating that the cases analyzed were representative of the whole population. The only significant difference was the NAC regimen used, reflecting the variability related to the inclusion period (Appendix A). The baseline characteristics of 126 patients with IHC data for all markers, according to the tumor clusters, are listed in Table 1. Median age was 66 (range, 61–72); 86% were pure urothelial carcinomas; the remaining 14% had mixed histology, with the predominance of squamous differentiation. Clinical staging was T2-4N0M0 for 104 (83%) patients; 21 (17%) had nodal involvement. The chemotherapy regimens used were: cisplatin and gemcitabine (CG) (63%), cisplatin, methotrexate, and vinblastine (CMV) (25%), and carboplatin and gemcitabine (CaG) (10%).

Pathologic down staging occurred in 55 (44%) patients; 39 cases (31%) had a pCR. The baseline characteristics of responders and non-responders were similar except for nodal involvement, which was more common among the non-responders (26% vs. 0%; *p* = 0.002) (Appendix A).

### 2.2. Taxonomic Classification and Association with Gene Mutations

We first compared the Nanostring RNA quantification of the four markers with protein expression using IHC. Highly significant Pearson correlation coefficients were found for GATA3, FOXA1, and KRT14; a good correlation was found for KRT5/6 (Figure 1A).

IHC analysis of FOXA1 and GATA3 showed a significant positive correlation (Pearson r = 0.68). KRT5/6 and KRT14 levels were also significantly positively correlated (r = 0.45). FOXA1 and GATA3 were negatively correlated with KRT14 (r= −0.53 and r = −0.62, respectively). GATA3, but not FOXA1, levels correlated negatively with KRT5/6 (r = −0.33). Overall, these findings are expected based on transcriptomics-based knowledge, thus validating the IHC-based tumor typing. Inter-core histoscore concordance analysis revealed a moderate (0.4–0.6) or substantial (>0.6) agreement for all markers. Concordance was higher for the BASQ group than for the luminal or mixed groups, emphasizing that IHC is a robust tool to classify BASQ tumors.

Because our aim was to test the suitability of IHC in select patients for NAC, we used the IHC-based histoscore to subtype the cases. To prevent the need of using thresholds, hierarchical clustering was used and identified three clusters. Cluster 1 (BASQ-like) (N = 47) is characterized by high KRT5/6 and KRT14 levels and low FOXA1 and GATA3 levels. Cluster 2 (urothelial/luminal-like) (N = 44) is characterized by high FOXA1 and GATA3 levels and low KRT5/6 and KRT14 levels. Cluster 3 (mixed-cluster) (N = 35) is characterized by a high expression of FOXA1, GATA3 and KRT5/6, and low KRT14 levels (Figure 2A). Re-clustering of bootstrapped protein expression showed that the “BASQ-like” cluster was the most stable (stability = 0.84), indicating that this group can be most robustly identified. Classification and regression trees (CART) analysis showed that the KRT14 and KRT5/6 levels provided the best cluster discrimination (Figure 2B). In agreement with global transcriptome analyses, two tumors were identified with a low expression of all markers, but they did not form a distinct cluster, likely due to the small number.

BASQ-like tumors were enriched for the subjects with a more advanced clinical stage at diagnosis (*p* = 0.013) and mixed squamous histology (*p* = 0.011), both findings in agreement with previous data and supporting the representativeness of the tumor series. In addition, BASQ-like tumors were associated with a higher number of resected nodes (*p* = 0.033). Importantly, patients with BASQ-like tumors had a higher rate of pCR to NAC (*p* = 0.034).

The first molecular taxonomy consensus classification did not include markers of luminal/urothelial differentiation [16]. Among the candidate molecules associated with urothelial differentiation are KRT20, a well established luminal marker, and *FGFR3, STAG2,* and *PIK3CA*—three major bladder cancer genes commonly altered in luminal tumors [22,23]. Therefore, we analyzed these markers in the same samples. KRT20 levels were significantly lower in the BASQ tumors (Appendix A) but it was similar in the other clusters. FGFR3 levels were significantly higher in the “mixed-cluster” tumors and the STAG2 levels were similar in all clusters. *FGFR3* and *PIK3CA* mutations were more common in the mixed-cluster tumors but differences did not reach statistical significance. *RAS* mutations were similarly distributed across clusters (Appendix A).

### 2.3. Association of Clusters with Response to NAC

To analyze the association with the response to NAC, we focused on the consensus markers of the BASQ-like cluster. The univariate analysis showed that the only factors significantly associated with the response to NAC were lymph node involvement (*p* = 0.011) and the BASQ-like cluster (*p* = 0.035). Of note, 16 patients (12.8%) had a pPR but only one of them had a BASQ-like tumor. The clinical variables and molecular factors associated with pCR are shown in Appendix A. The multivariate analysis of predictors of pCR, after adjusting for potential confounders, showed that the BASQ-like cluster (OR = 4.06; 95% CI 1.18–13.99) (*p* = 0.026) was significantly positively associated with pCR (Appendix A).

A few patients included in the study had a lymph node involvement or received carboplatin, features that are not part of standard NAC. When these cases were excluded (sensitivity analysis), the predictive value of the BASQ-like cluster was essentially unchanged (Appendix A). A fraction of the patients (24.6%) was treated with CMV combination therapy; while sample size did not allow stratified analyses, we found the same trend in the association of BASQ subtype with the outcome in this patient subgroup (Appendix A).

Mean follow-up in event-free patients was 85 months (range, 6–241); 50 patients (37.9%) progressed and 45 (34.1%) died from the disease. Figure 3 shows the Kaplan–Meier plots for relapse-free and disease-specific survival. Among all the patients, and among those who underwent a pCR, cluster classification was not associated with survival; patients with BASQ tumors having achieved a pCR had the best survival. By contrast, among the patients who did not have a pCR, those with BASQ tumors had the worst outcome (*p* = 0.11). These findings strongly suggest that the worse prognosis of patients with BASQ tumors is counteracted by a greater likelihood of displaying a pCR, resulting in the lack of differences in disease-specific survival.

### 2.4. Relevance to Clinical Implementation

Because the clinical implementation of the IHC classification of tumors would not rely on tissue microarrays (TMAs), we compared the marker expression in the TMAs (containing cores from the TURBT block) and in the full TURBT block sections from the corresponding tumors (n = 15). We found highly significant correlations for FOXA1 (r = 0.75), GATA3 (r = 0.78), and KRT14 (r = 0.83) and a modest correlation for KRT5/6 (r = 0.34) (Figure 1B). These correlations validate, overall, the usefulness of the TMAs for exploratory studies and agree with a recent publication [24].

Another level of heterogeneity relevant to clinical implementation is whether TURBT samples are representative of the deeper region of the tumor. To assess this, we compared the marker expression in full sections of paired TURBT and cystectomy specimens, from patients receiving NAC who did not achieve a pCR (n = 17), and in paired TURBT-cystectomy samples from patients from the same institutions who did not receive NAC (n = 15). In both sample series, the cystectomy samples showed a significantly reduced histoscore for luminal markers and a consistent - less prominent—increased levels of basal KRT markers. These results suggest heterogeneity in the cellular phenotypes in superficial vs. deeper tumor regions, with predominant luminal features in the TURBT samples (Appendix A). Interestingly, the trend of the changes in KRT14 levels was different in the samples that had been exposed, or not, to NAC suggesting an impact of the latter on tumor cell phenotype (Appendix A).

## 3. Discussion

Molecular knowledge has generated a plethora of hypotheses that could impact UBC patient management. The fact that only a fraction of patients respond to available therapies, highlights the need of improved patient stratification—especially with the advent of immunotherapy. This need is obvious for NAC, where the risk of tumor progression in non-responders has underscored its cautious application even when global survival statistics supports its use in patients with locally advanced organ-confined tumors [8].

In MIBC, several tumor subtypes have been identified; the greatest consensus exists for the BASQ-like type, defined using transcriptomics. Based on this, and on the increasing evidence that BASQ tumors are more aggressive but may respond better to chemotherapy, we performed this analysis using IHC.

To our knowledge, ours is the first study showing that a cluster of tumors with BASQ-like features, comprising approximately one-third of all patients with MIBC, can be robustly identified using simple IHC to predict the NAC response. This cluster is defined by the low/undetectable expression of GATA3/FOXA1 and the high expression of KRT5/6/14. KRT5/6 did not clearly distinguish BASQ-like tumors from the mixed-cluster tumor subtype. These markers have also previously been shown by the Lund group to robustly translate global transcriptomic classifiers into immunohistochemical assays [20]. Because of the well defined specificity of the antibodies used and the robustness of the IHC staining assays, the marker combination used here could be widely applied in the Pathology Departments of general hospitals where Nanostring technology is not available, thus facilitating the clinical translation. While it is possible that a lower number of markers could be used, at this point there is insufficient evidence for this: tumors that are broadly KRT14^+^ generally lack expression of GATA3 and FOXA1 and are unquestionably BASQ type; tumors that are KRT5/6^+^ and display a broad expression pattern are also highly likely to be BASQ-type, especially if they lack GATA3/FOXA1 expression; tumors with an expression of KRT5/6^+^ restricted to the basal layer are likely to be GATA3/FOXA1+. In these tumors, KRT5/6 distribution corresponds to that of the normal urothelium and therefore, they should not be considered BASQ like as highlighted by the Lund group [20].

Patients with BASQ-like tumors have a significantly higher likelihood of response to NAC in comparison to the other tumor subgroups, but only 42% of them underwent a pCR, indicating that further refinement is required to improve patient selection.

Three strategies have been applied to identify the markers of response to cisplatin-based NAC: global transcriptomics [15,18], panel IHC (first used here), and gene mutation analysis [25,26]. Two studies used global transcriptomics using formalin-fixed paraffin-embedded (FFPE) samples to define UBC subtypes in relationship to the response to NAC. Patients with basal tumors from a clinical trial with dose-dense MVAC and bevacizumab had improved survival compared to those with luminal or p53-like tumors. In that study, NAC response was not associated with tumor subtype [19]. In a large multicenter retrospective series comparing patients having undergone cystectomy without NAC and those having received NAC, the patients with basal tumors not receiving NAC consistently had worse overall survival than those with luminal tumors. Importantly, the patients with basal-like tumors (SCC-like, cluster III, or Uro B) showed the greatest improvement in outcome after NAC. In multivariate analysis, the patients with basal tumors not receiving NAC had an adjusted HR of 2.22 for OS compared to luminal tumors; these differences were eroded in the NAC cohort. Surprisingly, among the patients with basal tumors (unlike in luminal tumors), OS was similar for major responders vs. non-responders [18]. The use of group comparisons is complicated by the fact that some patients not undergoing NAC receive adjuvant chemotherapy—or eventually—receive cisplatin-based chemotherapy if they progress during follow-up.

Our study relied on the type of samples that would be used for patient stratification, i.e., TURBT. We find that a simple, robust, IHC-based typing of FFPE tumors may contribute to better select patients to receive NAC, although TURBT samples may not be completely representative of the whole tumor. The deep region thereof, sampled at cystectomy, tends to have a more basal phenotype, highlighting tumor heterogeneity. A recent study of bladder cancer histological variants has shown that BASQ tumors display the greatest heterogeneity, and supports that variants arise from conventional urothelial carcinomas [27]. More work needs to be carried out to specifically assess the relevance of sample bias, the difference between primary tumor vs. lymph node or distant metastases [25,28] as well as the biological changes associated with tumor evolution, naturally or under therapy pressure [26]. Prospective studies should determine the optimal strategy for BASQ tumor definition, considering the more recently proposed classifiers [29]. In this regard, it has been proposed that the "luminal-non-specified" subtype might also benefit from NAC. Since it is not currently possible to identify this group with IHC markers, this relevant question will have to be addressed in future studies either with RNA analyses or through improved IHC marker definition.

Regarding genomic alterations, somatic *ERCC2* mutations are enriched in NAC responders and functional in vitro studies support a causal association [30]. Genomic alterations in *RB1*, *ATM*, and *FANCC* were also predictive of a response to—and clinical benefit from—NAC [31]. The prevalence of mutations in these genes in MIBC is relatively low, and varies in different studies (*RB1* (20%, range 7–25%), *ATM* (11%, range 6–15%), *ERCC2* (10%, range 2–17%), *FANCC* (2.2%, range 1.5–3%)), and there is partial co-occurrence (source http://www.cbioportal.org/).

An important question is whether gene mutations are related to tumor subtypes, since this could impact the application of marker combination to predict the response to NAC. *ERCC2* mutation prevalence is similar in BASQ-like and non-BASQ-like tumors. *RB1* mutations—but not deletions—are significantly more common in BASQ-like than in urothelial tumors [9,29]. However, deletions are more common in the Lund “genomically unstable” subgroup. There is insufficient data to indicate whether *ATM* and *FANCC* alterations are enriched in any taxonomical types [9,11,29], suggesting that combined tumor subtyping and mutational analysis could improve the ability to predict NAC response.

This study has several limitations derived from its retrospective nature and not being a clinical trial, including the modest sample size and relatively long recruitment period, the lack of centralized pathology review, the inclusion of patients treated with carboplatin, possible uncontrolled differences in clinical management, and the generation of clusters based on TMA data. These limitations reflect much of the variability associated with the application of NAC in clinical practice [32,33] and have been mitigated through several types of analyses. While the P-values found are close to the threshold of significance, the OR are in the range of clinical relevance. Nevertheless, the establishment of robust thresholds for tumor classification and the conduct of prospective clinical studies are necessary to conclusively establish the contribution of tumor subtyping to the management of patients with MIBC and the optimal clinical implementation strategy.

## 4. Materials and Methods

### 4.1. Study Design

This study included patients treated at two Spanish hospitals (Institut Català d´Oncologia, Hospital Universitari Germans Trias i Pujol (HUGTiP), Badalona; Fundació Althaia, Manresa, Barcelona). The stuy protocols were approved by the corresponding ethical committees [codes PI-15-072 (HGTiP) and 15-27 (Fundació Althaia)].MIBC was identified by the transurethral resection of the bladder tumor (TURBT). Patients clinically staged with abdominal/pelvic computed tomography (CT) and a chest X-ray, and classified as T2-4aN0-2M0, were candidates for cystectomy after NAC. pCR was defined as the absence of a detectable tumor in the cystectomy specimen (pT0N0); partial response was defined as down staging to non-MIBC (<pT2N0). Remaining cases were considered non-responders. None of the patients received adjuvant chemotherapy nor immunotherapy. Patients provided informed consent; the study was approved by the institutional review board of the two hospitals.

NAC was administered to 215 patients (1994–2014). The chemotherapy regimen was cisplatin, methotrexate, and vinblastine (CMV) until 2000, after which cisplatin and gemcitabine (CG) or—exceptionally—carboplatin and gemcitabine (CaG) was used. The final number of patients analyzed was 126 patients; the remaining patients were mainly excluded because the tumor tissue was unavailable or information on all four IHC markers was not obtained. Information was collected through a retrospective review of clinical and pathological records.

### 4.2. Immunohistochemical Analyses

TURBT formalin-fixed paraffin-embedded (FFPE) blocks were reviewed. Representative tumor areas were selected to extract 3 cores and construct tissue microarrays (TMAs). Cores were positioned non-consecutively to avoid artefacts; TMAs were constructed following the established guidelines and sectioned; and the slides were embedded in paraffin and stored at 4 °C. The following antibodies were used: KRT5/6 (PRB-160P, Covance; 1/2000), KRT14 (PRB-155P, Covance, 1/2000), GATA3 (CM405 A, Biocare Medical, 1/300), FOXA1 (ab170933, Abcam, 1/100), FGFR3 (B9, Santa Cruz), KRT20 (Ks20.8, Dako), and STAG2 (J-12, Santa Cruz). After deparaffinization, antigen retrieval, and endogenous peroxidase blockade, the sections were incubated with antibodies overnight, washed, and Envision secondary reagents (Agilent) were added for 1 h. After washing, the reactions were developed with DAB, the sections were counterstained with hematoxylin, and mounted. Scoring was performed blind to clinical/pathological information by an experienced investigator (FXR). The proportion of reactive cells (0–100%) and staining intensity (0–3) were recorded; the histoscore was calculated as the product: Intensity x % reactive cells (range 0–300). The average of the scores was used for analysis.

### 4.3. RNA Expression Analyses

Total RNA was isolated from the full FFPE sections and purified from macrodissected high-density tumor areas using the truxTrac FFPE RNA microTube kit (Covaris, Woburn, MA, USA). Cell lysates were sheared by sonication; RNA was eluted and quantified using Qubit. Gene expression analysis was conducted on the NanoString nCounter platform. Samples were scanned at maximum resolution on the nCounter Digital Analyzer. Four housekeeping transcripts were used for normalization (*GAPDH*, *ACTB*, *HPRT1*, and *LDHA*).

### 4.4. Mutational Analyses

Hotspot mutations in *FGFR3* (R248C, S249C, G372C, S373C, Y375C, G382R, A393E, K652E/Q/T/M), *PIK3CA* (E542K, E545K/Q/G, H1047L/R), *HRAS* (G12C/S/D/V, G13C/R, Q61K/L/R), *KRAS* (G12C/R/S/A/D/V, Q61E), and *NRAS* (G12/R, Q61/L/R) were assessed. DNA was extracted from the tumor cell-enriched areas using a DNeasy tissue kit (Qiagen, Hilde, Germany). Multiplex PCRs were performed; the products were analyzed with the a SNaPshot Multiplex Kit (Applied Biosystems) [22].

### 4.5. Statistical Methods

All the analyses performed were specified a priori and corresponded to specific hypotheses. Variables were summarized as the means, medians, and standard deviations—when continuous—and as percentages when categorical. Bivariable associations were evaluated by the chi2 test (categorical) and the Kruskal–Wallis test (continuous). The Pearson test was applied to assess the correlations. Agglomerative hierarchical clustering was applied using the Ward's minimum variance method to investigate the evidence for the natural groupings of tumor (subtypes) based on the correlations between expression profiles. Histoscores were standardized to a 0–1 range and considered as input. Heatmaps were generated using pHeatmap [34]. To investigate cluster stability, bootstrapped histoscore values were re-clustered and their similarity with the original classification was evaluated with the Jaccard coefficient [35]. The CART model was trained considering the 3 clusters and the 4 markers using the caret and rpart packages. Leave-one-out cross validation was used to evaluate the best prediction scenario.

Adjusted logistic regression was applied to assess the independent association between the clusters and response. For each variable, the odds ratio (OR) was estimated referring to the association between the predictor pCR with the ‘no-complete response’ group—including partial responders and non-responders—as the reference. Sensitivity analyses were run after excluding subjects with pelvic lymph node involvement or treated with carboplatin (N = 31).

Survival curves were derived using Kaplan–Meier methods and compared with log-rank tests by strata defined by the tumor subtypes obtained with the cluster analysis. Disease-specific survival and progression-free survival were considered as outcomes. Hazard ratios and 95% confidence interval for the association between the subtypes and the outcomes were estimated with univariable and multivariable Cox regression models.

Differences were considered statistically significant at two-sided *p* < 0.05. All statistical analyses were performed using R version 3.3 (https://www.r-statistics.com/tag/r-3-3-0/) unless other otherwise indicated. Data availability statement: All data from this study are available upon reasonable request to authors.

## 5. Conclusions

This retrospective study shows that patients with BASQ-like tumors can be identified using immunohistochemistry on paraffin-embedded tissue and are 4-fold more likely to achieve a pathological complete response to platinum-based NAC. The disease-specific survival of patients with BASQ-like tumors treated with NAC was not different from that of other tumor subtypes. Prospective studies are required to confirm these observations and determine the clinical value of tumor subtyping.

## Figures and Tables

**Figure 1 cancers-12-01784-f001:**
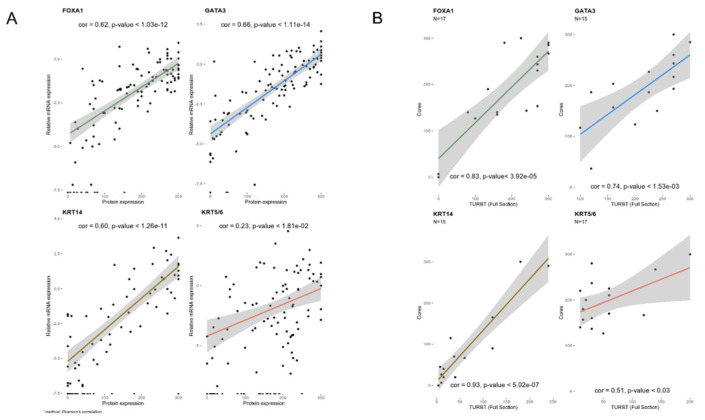
(**A**) Comparison of the results of immunohistochemistry (IHC) findings and Nanostring expression for the markers used for the tumor subtype classification shows an excellent correlation between both assays. (**B**) Comparison of the histoscore of tissue microarray (TMA) cores with the findings in full sections of the corresponding transurethral resection of the bladder tumor (TURBT) full block sections.

**Figure 2 cancers-12-01784-f002:**
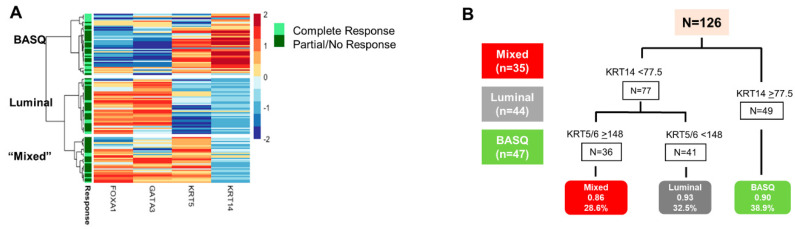
Expression of FOXA1, GATA3, KRT5/6 and KRT14 in the three tumor subtypes. The heatmap depicts the relative biomarker expression. (**A**) Clusters resulting from the KRT5/6, KRT14, GATA3, and FOXA1 tumor histoscore. (**B**) Classification and regression trees (CART) model showing the optimal accuracy of the cluster classification. The numbers in the bottom boxes are the proportion of the observed clusters predicted and the percentage of UBC s each class represents.

**Figure 3 cancers-12-01784-f003:**
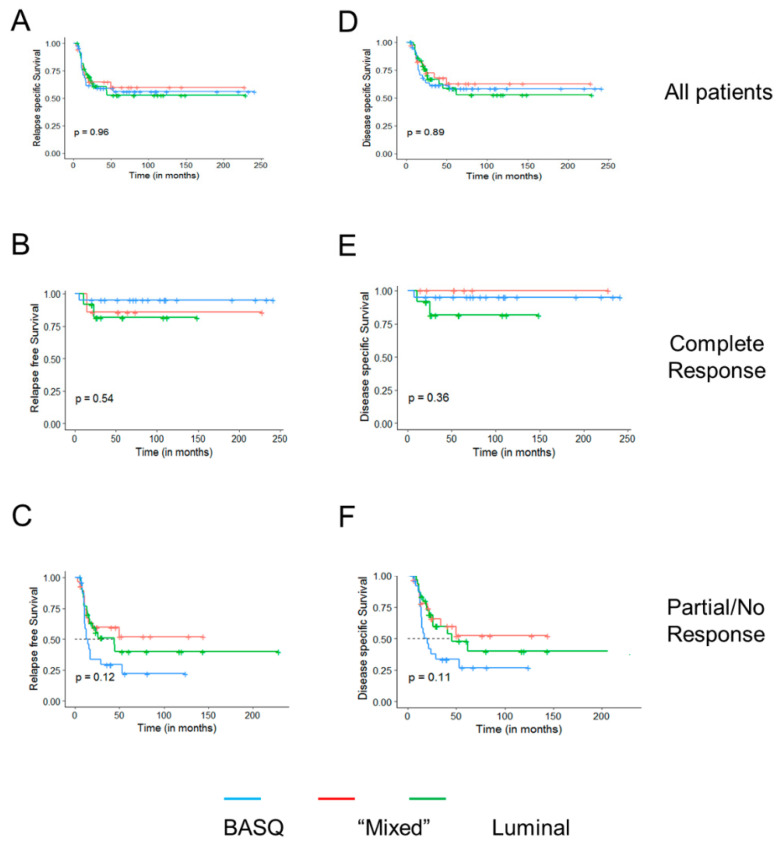
Kaplan–Meier relapse-free survival (**A–C**) and disease-specific survival (**D–F**) curves for all patients (**A**,**D**), patients undergoing a pathological complete response (pCR) (**B**,**E**), or partial/non-responders (**C**,**F**) according to the three tumor subtypes.

**Table 1 cancers-12-01784-t001:** Patient and tumor characteristics according to the taxonomic clusters (N = 126).

Characteristics	Total N = 126	Basq_Like N = 47	Luminal_Like N = 44	Mixed N = 35	*p* value
Sex					1.000
Male	118 (93.7%)	44 (93.6%)	41 (93.2%)	33 (94.3%)	
Female	8 (6.3%)	3 (6.4%)	3 (6.8%)	2 (5.7%)	
Age (Med, IQR)	66 (61–72)	66.0 (62.0;72.0)	67.0 (62.0;72.2)	65.0 (57.5;71.5)	0.481
Morphology					0.011
Urothelial (pure)	108 (85.7%)	33 (70.2%)	43 (97.7%)	32 (91.4%)	
Mixed (squamous component)	14 (11.1%)	10 (21.3%)	1 (2.27%)	3 (8.57%)	
Mixed (adenoca. component)	2 (1.6%)	2 (4.26%)	0 (0.00%)	0 (0.00%)	
Other*	2 (1.6%)	2 (4.26%)	0 (0.00%)	0 (0.00%)	
Grade					0.967
Low	3 (2.4%)	1 (2.13%)	1 (2.27%)	1 (2.86%)	
High	123 (97.6%)	46 (97.9%)	43 (97.7%)	34 (97.1%)	
Lymphovascular invasion					0.576
No	97 (77.0%)	39 (83.0%)	34 (77.3%)	24 (68.6%)	
Yes	24 (19.0%)	7 (14.9%)	9 (20.5%)	8 (22.9%)	
'Missing'	5 (3.0%)	1 (2.13%)	2 (4.5%)	3 (8.6%)	
Hydronephrosis					0.749
No	77 (61.1%)	27 (57.4%)	27 (61.4%)	23 (65.7%)	
Yes	49 (38.9%)	20 (42.6%)	17 (38.6%)	12 (34.3%)	
cTNM					0.017
T2N0	11 (8.80%)	2 (4.35%)	9 (20.5%)	0 (0.00%)	
T3/4N0	93 (74.4%)	37 (80.4%)	27 (61.4%)	29 (82.9%)	
TxN1M0	21 (16.8%)	7 (15.2%)	8 (18.2%)	6 (17.1%)	
'Missing'	1 (0.8%)	1 (2.1%)	0 (0.00%)	0 (0.00%)	
NAC treatment					
CG/CaG	91 (72.2)	32 (68.1)	35 (79.5)	24 (68.6)	0.014
CMV	31 (24.6)	15 (31.9)	5 (11.4)	11 (31.4)	
Other*	4 (3.2)	0	4 (9.1)	0	
pTNM					0.034
Complete	39 (31.2%)	20 (42.6%)	12 (27.3%)	7 (20.0%)	
Partial	16 (12.8%)	1 (2.1%)	8 (18.2%)	7 (20.0%)	
Not responder	70 (56%)	26 (55.3%)	23 (52.3%)	21 (60.0%)	
'Missing'	1 (0.79)	0 (0.00%)	1 (2.27%)	0 (0.00%)	
Lymph node invasion					0.553
No	97 (80.2%)	39 (84.8%)	34 (79.1%)	24 (75.0%)	
Yes	24 (19.8%)	7 (15.2%)	9 (20.9%)	8 (25.0%)	
'Missing'		1 (2.13%)	1 (2.27%)	3 (8.57%)	
Lymph nodes resected					0.033
0–9	67 (55.4%)	29 (63.0%)	17 (39.5%)	21 (65.6%)	
10+	54 (44.6%)	17 (37.0%)	26 (60.5%)	11 (34.4%)	
'Missing'	5 (4%)	1 (2.13%)	1 (2.27%)	3 (8.57%)	

TNM: tumor node metastasis classification; CG: cisplatin and gemcitabine; CMV: cisplatin, methotrexate and vinblastine; CaG: Carboplatin and gemcitabine; MVAC: Methotrexate, vinblastine, adriamycin, and cisplatin; *Other: dose-dense MVAC.

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
