# Peer review of "Immunohistochemistry-Based Taxonomical Classification of Bladder Cancer Predicts Response to Neoadjuvant Chemotherapy"

_cancers, 2020, doi:10.3390/cancers12071784_

Round 1

Reviewer 1 Report

The authors classified 126 cases of post-neoadjuvant urinary bladder urothelial carcinomas by using IHC (FOXA1, GATA3, KRT5/6/14) into 3 clusters (basosquamous-like, luminal-like, mixed), and correlated the clusters with RNA expression and mutations. They found that BASQ-like tumours were more likely to achieve a pathological response to
neo-adjuvant chemotherapy. It is a well conducted study with a moderate sample size. This molecular classification has been a topic of great interest currently.

The authors employed hierarchical clustering instead of thresholds to identify the clusters. However, for practical use, thresholds are more commonly used. For example, how do we classify a case by using the IHC in individual cases? As shown in Figure 2B, it seems that using KRT14 and KRT5/6 alone can successfully classify the cases. If it is true, I recommend the authors to provide a practical guideline for readers who want to use the same classification for their cases.

The case numbers of each clusters obtained from CART (figure 2B) were not the same as the numbers obtained from hierachical clustering (Figure 2A). I wonder which one was more precise in predicting outcomes. Did the authors do the same statistical analyses using the classification in Figure 2B?

The middle numbers in the bottom boxes of Figure 2B (0.86, 0.93, 0.90) were not explained (r?). They should be labeled in the figure legend.

Reviewer 2 Report

General

The authors seek to verify the previous findings that a particular molecular subtype of bladder cancer had better response to neoadjuvant therapy in their own cohort of 126 muscle-invasive bladder cancer. Overall the paper is well written. The data is a useful resource if the authors could share it as Suppl. Material or deposit the data into public repository such as GEO would be great. However, in its current form, I’m not convinced the study offer additional understanding to the disease but merely a verification of what has been reported. The authors may put emphasis and more in-depth analysis on their findings.

For example, the authors mentioned the study contribution including (1) studying IHC of luminal/urothelial differentiation markers however the data is not shown and is not used in tumor subtyping. Why is that? (2) The mutation data were described by two sentences in the result with limited details: hotspot? What type of mutation?

In my view, the authors may want to focus on the data comparing TURBT vs Cystectomy as this is relative rare. The authors could assess the tumor heterogeneity and investigate if there is difference in subtype in pCR, pPR and pNR cases, and if heterogeneity is observed in mutation. The subtype from TURBT and from cystectomy, which one would have better association to response to NAC.

Specific:

Since there are mainly two regimens for the NAC, the association analyses should be repeated for each regimen.

There have been updates to the molecular subtype for example, https://pubmed.ncbi.nlm.nih.gov/30213523/, https://pubmed.ncbi.nlm.nih.gov/31563503/, https://clincancerres.aacrjournals.org/content/early/2018/09/15/1078-0432.CCR-18-1106. In particular, a group of bladder cancer sample of luminal subtype (luminal-non-specified) was found to be benefited from neoadjuvant therapy as well.

Please give the reason of why 76 and 7 samples were excluded.

I would prefer the partial response be separated from no response, as consistent with clinical practice. What is the number of partial response and no response?

It is interesting that KRT5/6 has little mRNA expression but high protein level, why is that? This is important as KRT5/6 is said to be the best discriminating feature for the different subtype/cluster

Results 2.2, the authors mentioned using Spearman correlation for correlation analysis. However, in the text after figure 1 the results were described using Pearson correlation, and that’s confusing. Which test was used and why? While I appreciate the authors effort in validating the correlation and IHC and the concordance of tumor typing using mRNA and protein, such is a repeat from Sjodahl et al., 2016. Thus, I would like to suggest the authors to consider putting the correlation analyses into supplementary and go straight to hierarchical clustering of IHC.

Results 2.3, in strict sense, the goals of prediction and of association assessments are different, hence, the methods are usually different. For predictor, I would expect a predictive model ready to be validated on another cohort. Here, the authors seek to assess the parameters associated with pCR. I suggest the authors to rephrase. Having a table legend for suppl. table 5 to explain what is model 1 and model 2 would be great.

Presentation:

  • A few representative slides for the different subtypes and markers would be beneficial to convince the reader that the antibody is really working. The scoring was performed by one pathologist or by image processing software?
  • The authors may want to use mRNA expression and protein abundance/level to facilitate reading so the readers know when the authors are talking about protein/IHC and when are about mRNA.
  • Fig. 1A a log-2 scale for NanoString relative expression may look better. Please explain relative expression in figure legend.
  • Fig. 2A, label “KRT5” should be changed to “KRT5/6” for consistency with the text and other figures.
  • A boxplot would be better for showing the IHC levels instead of suppl. Table 3.
  • Fig. 3 figure legend for D, E, F missing.

Round 2

Reviewer 2 Report

i would prefer the authors include some of the reply to reviewer in the paper for "completeness". Specifically,

1

"We feel that it is worthwhile emphasizing this point because the previous work referred to all TURB samples - regardless of patients receiving subsequently NAC. In contrast, our analyses focus on patients going on to receive NAC and there might be differences between both sample types"

please put this emphasis in the text to highlight the difference with the Sjodahl's work.

2.

"The main reason for lack of tissue availability (n=76) was that one hospital is a referral centre for other hospitals in its catchment area and the TURBT had been performed elsewhere. However, since we are concerned about possible selection biases, we have compared clinical-pathological characteristics between the groups of eligible vs ineligible patients and found no significant differences (Supplementary Table 1). As discussed in the text, we did not find strong evidence for the existence of major biases. For the other 7 patients, the reason of exclusion was lack of information on all markers required for cluster definition.  "

what about include this explanation in the patient inclusion/exclusion part in M&M because as a reader i have this question.

3.

"We considered that the markers used have been so extensively investigated using IHC that it was not necessary to include color figures to support the scores. The latter were visually estimated, without the use of image processing, by an experienced investigator."

please include "by an experience investigator (initial)" in the IHC section in M&M.

4. "we have now analyzed separately the group of patients receiving CG/CaG vs. CMV"

Since the authors mention the result, the figure should be in the supplementary fig.

Author Response

We have now responded to all requests, as per attached file.
